# Gender-related and non-gender-related female homicide in Porto Alegre, Brazil, from 2010 to 2016

**Angelita Maria Ferreira Machado Rios**[1,2]**, Kleber Cardoso Crespo**[1,2]**, Murilo Martini**[1]**, Lisieux Elaine De Borba Telles**[1]**, Pedro V. S. Magalhães**[1] *

**1** Hospital de Clínicas de Porto Alegre, Clinical Research Center, Graduate Program in Psychiatry and Behavioral Sciences, Universidade Federal do Rio Grande do Sul, Porto Alegre, Brazil, **2** Porto Alegre Medicolegal Department, Instituto Geral de Perícias do Rio Grande do Sul, Porto Alegre, Brazil

\* pmagalhaes@hcpa.edu.br

## Abstract

Female homicide is a global phenomenon with a higher prevalence in countries in Asia, Africa, and the Americas. Life expectancy in Brazil is compromised by the high risk of death from male and female homicides, a growing social problem. This study aimed to integrate different public datasets and describe the sociodemographic, criminal, and medicolegal characteristics of the homicides of girls and women occurring in Porto Alegre, southern Brazil, from 2010 to 2016. The data were obtained from autopsy reports, police reports, and records from crime scenes. During this period, there was a significant increase in overall standardized rates of female homicides (4.98 to 10.85), with a pronounced increase in non-gender-related deaths, especially due to urban violence, such as involvement in drug trafficking and other crimes and robbery resulting in death. Young (15–29 years of age), non-White women were the most affected. Increased female homicide rates due to non-gender-related factors is a new and worrying phenomenon in Brazil. Obtaining specific data on the profile of victims and characteristics of violence is a crucial step in facing the problem and directing public policies.

## Introduction

Homicide rates are considered the best international indicator of violence in a country [1]. In recent decades, homicide mortality trends have varied across different regions of the planet. While several regions show a downward trend, Latin America seems to be a special case, with a sustained increase [2]. Significant elevations have been observed in this region since the 1950s, affecting both sexes. Male homicides are higher in number and are generally committed by people who are strangers to the victim (outside the family sphere), while women are more frequently killed by people they know [3].

The incorporation of feminicide or femicide into the criminal code of several countries represented an advance in gender equality policies, but this alone was not sufficient to reduce gender-related homicide rates and to minimize the complex cultural, social, and economic

identification of victims of violent crimes. Unidentified data may be made available upon request to Hospital de Clínicas de Porto Alegre Ethics Committe at cep@hcpa.edu.br.

**Funding:** This study was funded by Conselho Nacional de Desenvolvimento Científico e Tecnológico (304184), Coordenação de Aperfeiçoamento de Pessoal de Nível Superior (001), and Hospital de Clínicas de Porto Alegre (2016).

**Competing interests:** The authors have declared that no competing interests exist.

influences on violence against women [3]. In 2017, 87,000 female homicides were reported worldwide; 58% were committed by partners or other family members (50,000 homicides or 1.3 homicides/100,000 women) and over a third of the deaths (30,000) were caused by intimate partners. Globally, 137 women were murdered daily by a member of their own family. The greatest number of deaths was in Asia (20,000 women), followed by Africa (19,000 women) and the Americas (8,000 women) [3].

In Latin America, high overall homicide mortality rates are related to high unemployment rates, participation in criminal activities, and easy access to firearms [4, 5]. There is also a higher concentration of deaths among women aged 15 to 29 years [6]. In Brazil, life expectancy is compromised by the high risk of death from male and female homicides [5]. A study that analyzed homicide mortality in women aged 10 years or over in different Brazilian regions from 1980 to 2014 revealed an average rate of 5.13 deaths per 100,000 women. In all Brazilian regions, younger women were at a higher risk of dying from homicide, with an upward trend of mortality rates for women born in 2000–2004 [7]. This violence has enormous impacts, such as physical and psychological suffering of family members, socioeconomic losses, and legal and prison system effects [8–11].

Several risk factors have been identified for the violent and premature death of women: young age, termination of the romantic relationship by the woman, change of partners, low level of education, unpaid occupation or insufficient income, previous violence in the relationship or during pregnancy, consumption of alcohol/drugs by the offender/victim or both, and easy access to firearms [12–14]. The increase in female homicides has challenged the classic criminal studies of the second half of the 20th century conducted by Veli Verkko (1967), who suggested that fluctuations in homicide rates were based on the increase or decrease in male homicides while female homicides remained stable. This phenomenon was attributed to the woman living in a more peaceful atmosphere than the man [15]. However, with greater female participation in social activities, this started to change, including the context of female victimization. With modifications in the role and status of women in society, gender violence was then attributed to situational factors, which were related to professional risks and greater social exposure, but also to motivational factors arising from domestic violence, generally inflicted by the intimate partner in response to changes in power and dominance within the household. Over the years, women continued to play new roles and compete for positions in society that had male characteristics and prerogatives, including illicit activities such as organized crime. Studies addressing violent deaths of women described the hostility of both the external and domestic environments, with a scientific and political interest in research on female victimization [16, 17].

Violent deaths of women have been recognized as a growing social problem in Brazil. Nonetheless, specific and qualified data on the profile of these victims are limited by the small number of studies involving public safety variables and the challenge of reconciling data from different public databases. Thus, this study aimed to describe the characteristics of the homicides of girls and women occurring from January 2010 to December 2016 whose bodies were examined at the morgue of the Medicolegal Department of Porto Alegre, southern Brazil, with an analysis of sociodemographic, criminal, and medicolegal variables of the victims. We report global changes in homicide rates in women in the period, as well as changes in relevant subgroups according to age, race and motivation for the homicide.

## Materials and methods

This retrospective study of female homicides occurring from January 2010 to December 2016 was based on an analysis of autopsy reports issued by the Porto Alegre Medicolegal

Department. The estimated average population covered by the morgue in the study period was 2,334,730 people, including the population of the Rio Grande do Sul state capital, Porto Alegre (60.8%), and nine additional cities in the metropolitan area.

Female deaths classified as homicides were included in the study. Violent deaths due to accidents or undetermined causes were excluded. This sample (ten central cities) was chosen because forensic experts attend the crime scene in all cases of violent death, which produces a larger and more accurate source of information about victims and offenders, crime circumstances, and motive for the criminal act. Body transportation in these ten cities is carried out by forensic technicians, ensuring the chain of custody and information security.

To study the phenomenon of violent and premature death of women, sociodemographic (victim's age, skin color, and city where the death occurred), criminal (history of previous victimization), and medicolegal variables (agent or instrument causing the death, site of death and number of injuries, and presence of alcohol and/or psychotropic drugs in the body) were considered. Sociodemographic and criminal information was obtained from the Forensic Institute database, police reports, and records collected at the crime scene by the removal team; data regarding prior victimization were obtained from existing prior records of the Forensic Institute. Medicolegal variables were extracted from autopsy reports and forensic laboratory test results. Skin color (classified as White or non-White) was ascertained on postmortem examination, together with the number of injuries and causative agent.

Police reports and forensic crime scene analyses were used to make an initial hypothesis of motive for the crime. Homicides were then classified as those perpetrated by an intimate partner (femicide), by another family member (family-related homicide), sex crimes, homicides related to criminal activities (eg, drug trafficking and other crimes), and robbery resulting in death. The cases could then be as gender-related homicides (family-related including femicide and sex crimes) and non-gender-related homicides (robbery resulting in death and those related to criminal activity) to follow the United Nations Office on Drugs and Crime (UNODC) classification (2018) and to compare them with international data (see Table 1). The classifications were made by two authors (AR and KC) belonging to the Institute's staff with extensive experience in forensic psychiatry. Liver or urine samples were analyzed for the presence of drugs, and blood samples were used for the presence of alcohol on autopsy

**Table 1. Definitions, criteria used and examples of police reports on female homicide.**

| Motivation | Source | Characteristic | Criteria | Sample report |
|---|---|---|---|---|
| **Femicide** | Intimate partner | Gender-related* | Police notification or crime scene assessment showing partner as suspect | *Woman's body at the crime scene and partner injured from attempted suicide* |
| **Family-related** | Other family member | Gender-related | Police notification or crime scene assessment evidencing crime-motivating family quarrel | *Son assaulted the mother with a knife inside the residence* |
| **Sex crime** | Non-family related perpetrator | Gender-related | Police notification or crime scene assessment suggesting sexual violence | *Corpse found in isolated place, underwear removed, presence of signs suggestive of sexual violence* |
| **Drug traffic related** | Urban violence | Non-gender-related | Police notification suggesting criminal faction action or modus operandi of crimes related to drug trafficking | *Home invasion with execution of members of factions* |
| **Other crime related** | Urban violence | Non-gender-related | Police notification or site assessment reporting conflicts unrelated to the family environment or drug trafficking | *Previous disagreement with a neighbor motivating the murder* |
| **Robbery resulting in death** | Urban violence | Non-gender-related | Police notification or site assessment reporting crime dynamics | *Victim parked vehicle and was approached by criminals announcing assault* |

* The definition of gender-related homicide follows the UNODC report "Gender-related killing of women and girls" and analyzed using the indication of partner and family-related homicide

examination. Toxicology was performed by semiquantitative enzyme immunoassay until 2015 and by immunochromatography from then on. The Headspace method was used for alcohol.

Chi-square tests were used to test differences between groups. A logistic regression model constructed a priori and consisting of age, skin color, marital status, previous occurrences, place of death, the occurrence of other deaths in the same place, the instrument used, single injury, and toxicology was used to predict gender-related homicides. Homicide rates were calculated according to age group, skin color, and type of homicide. Resident population, stratified by sex, age, year and skin color were obtained from the Brazilian Institute of Geography and Statistics (IBGE). For the calculation of age-standardized rates, the number of deaths in the analyzed year was considered as the numerator and the estimated population of the respective year was considered as the denominator.

Rate variations per year and comparison of trends between groups were made by joinpoint regression using NCI software (18). Joinpoint regression is an increasingly popular method for analyzing trends in time series data as it can help identify calendar years in which the temporal trend changed significantly. The method assumes that data can be divided into subsets with unique linear trends. A joinpoint is a point in time when the population parameters change. Joinpoint regression joins multiple straight lines in logarithmic scale in order to detect the annual trend. The analysis starts with the minimum number of inflections to assess whether one or more joinpoints are statistically significant and whether they should be added to the model. For the analysis of trends, we sought to identify the regression equation that best described the relationship between the independent variable (year) and the dependent variable (homicide rates). We calculated rates and standard errors for each year for each group of interest and thus obtained the annual percent change (APC) for each group.

The study was authorized by the Division of Education and Research of the Porto Alegre Medicolegal Department and approved by the Research Ethics Committee of Hospital de Clínicas de Porto Alegre (project no. 899062). The ethics committee waived the requirement for informed consent from next of kin.

## Results

From January 2010 to December 2016, 6,981 homicides were referred to the Porto Alegre Medicolegal Department, of which 486 (6.5% of the total) were female deaths. The youngest female victim was 1 day old and the oldest was 89 years old, with a median age of 29 (IQR 21–40) years (Table 2). There was a significant increase in female homicide rates in the study area from 2010 to 2016 (Table 3). The overall standardized rate rose from 4.98 to 10.85 (average APC [AAPC] 7.1%, 95% confidence interval [CI] 7.6–12.8%), with a significant joinpoint in 2013. Among the subgroups by age, the 15-19-year-old group behaved similarly to that of overall deaths, with a significant change after 2013, when it reached the highest rates in the study period.

Average APC reported due to significant joinpoint present

In 5.6% of cases, it was not possible to determine the type, perpetrator, or motive of the homicide from the available information. Of the remaining cases (459), 73.2% were linked to criminal activity (55.3% drug trafficking, 10.9% other crimes, 7% robbery resulting in death), 24.4% involved the family sphere (20.3% femicides and 4.1% family-related crimes), and 2.4% were sex crimes. They were also grouped into gender-related (26.8%) and non-gender-related (73.2%) cases (Fig 1). There were important differences in the characteristics of the 4 types of homicides, summarized in Table 1. While sex crimes affected the younger groups, those related to criminal activity were reported in the intermediate age groups, and robbery resulting in death occurred in the older groups. In a logistic regression model (AUC 0.84, Nagelkerke

**Table 2. Characteristics of homicide victims from the Porto Alegre Medicolegal Department from January 2010 to December 2016 according to motive (n = 459).**

| | | Family-related (n = 112) | Sex crime (n = 11) | Robbery resulting in death (n = 32) | Criminal activity (n = 304) | Total (n = 459) |
|---|---|---|---|---|---|---|
| Porto Alegre* | | 50.9% | 63.6% | 75.0% | 62.8% | 60.8% |
| Age range* | <15 | 4.5% | 18.2% | 0.0% | 3.3% | 3.7% |
| | 15–34 | 58.0% | 72.7% | 21.9% | 67.4% | 62.1% |
| | 35–64 | 33.9% | 9.1% | 65.6% | 28.0% | 31.6% |
| | >64 | 3.6% | 0.0% | 12.5% | 1.3% | 2.6% |
| Non-white skin color* | | 22.3% | 36.4% | 9.4% | 34.3% | 29.7% |
| Marriage status* | Single | 71.7% | 100.0% | 53.1% | 87.5% | 81.5% |
| | Married | 26.4% | 0.0% | 40.6% | 11.8% | 17.1% |
| | Widow | 1.9% | 0.0% | 6.3% | .7% | 1.4% |
| Previous physical victimization | | 64.3% | 72.7% | 53.1% | 73.0% | 69.5% |
| Previous emotional victimization | | 80.4% | 81.8% | 81.3% | 86.5% | 84.5% |
| Previous sexual victimization | | 8.9% | 27.3% | 6.3% | 11.2% | 10.7% |
| Multiple victims | | 30.6% | 0.0% | 23.3% | 38.6% | 34.6% |
| Death at home* | | 72.9% | 0.0% | 38.7% | 23.6% | 36.0% |
| Instrument* | Firearm | 42.9% | 9.1% | 62.5% | 87.5% | 73.0% |
| | Knife | 37.5% | 27.3% | 28.1% | 4.9% | 15.0% |
| | Fall | 8.0% | 18.2% | 6.3% | 4.3% | 5.7% |
| | Asphyxiation | 9.8% | 45.5% | 3.1% | 2.6% | 5.4% |
| | Immolation | 1.8% | 0.0% | 0.0% | .7% | .9% |
| Single lesion* | | 42.0% | 57.1% | 50.0% | 28.8% | 33.8% |
| Toxicology present for alcohol | | 21.6% | 10.0% | 8.3% | 22.0% | 20.8% |
| Toxicology present for other drugs* | | 10.4% | 20.0% | 4.0% | 40.5% | 30.3% |

*Chi-square statistic significant at the 0.05 significance level. Indicates crime frequency is over-represented within the same line at the 0.05 significance level

R2 = 0.42), significant predictors of gender-related homicide were age (OR 0.97, 95% CI 0.94–0.99, p = 0.011), death at home (OR 7.70, 95% CI 3.88–15.30, p<0.001), single injury (OR 6.46, 95% CI 1.22–4.57, p = 0.011), no drugs of abuse in the toxicology test (OR 3.20, 95% CI 1.39–7.37, p = 0.006), and use of methods other than firearms (OR 0.14, 95% CI 0.07–0.29, p<0.001). Hosmer & Lemeshow test indicated a good fit for the model (p = 0.247).

Rates changed significantly according to the type of homicide. Non-gender-related homicide had an upward trend (AAPC 11.40) while gender-related homicide remained stable (AAPC 2.34). Skin color was also involved, with higher rates in non-White women (Fig 2).

## Discussion

The study showed a significant increase in the female homicide rate in the study area from 2010 to 2016, which followed closely the change in male homicide in this population. Although limited geographically, by separating the type of homicide by motive, we demonstrated increased rates due to non-gender-related crimes, linked to criminal activities and robbery resulting in death. Gender-related homicide rates remained stable over the study period.

Assessing changes in female homicide rates according to the type of crime can help understand the phenomenon. In his dynamic laws, Verkko described that changes in this type of violence predominantly affect male victims, which has been confirmed in recent studies [15, 16]. In this study, however, we demonstrated that changes in homicide rates over the study period

**Table 3. Female homicide rates from the Porto Alegre Medicolegal Department from January 2010 to December 2016 per 100,000 population.**

|  |  | 2010 | 2011 | 2012 | 2013 | 2014 | 2015 | 2016 | AAPC |
|---|---|---|---|---|---|---|---|---|---|
| **Age range** |  |  |  |  |  |  |  |  |  |
|  | 0–4 | 4.21 | 2.81 | 1.38 | 0.00 | 0.00 | 1.32 | 1.31 | -7.90 |
|  | 5–9 | 0.00 | 0.00 | 0.00 | 0.00 | 0.00 | 4.34 | 0.00 | **152.65***  |
|  | 10–14 | 1.11 | 0.00 | 0.00 | 1.19 | 1.23 | 1.28 | 2.68 | 88.90 |
|  | 15–19 | 10.04 | 12.20 | 8.80 | 5.45 | 8.73 | 16.47 | 25.66 | **12.50***  |
|  | 20–24 | 18.69 | 7.33 | 18.92 | 11.59 | 8.50 | 7.53 | 17.27 | -4.30 |
|  | 25–29 | 7.60 | 10.52 | 8.72 | 7.87 | 13.03 | 10.22 | 12.48 | **6.00***  |
|  | 30–34 | 2.08 | 7.09 | 7.91 | 2.91 | 13.42 | 15.42 | 14.63 | **26.70***  |
|  | 35–39 | 8.46 | 4.75 | 6.92 | 7.81 | 10.87 | 7.45 | 4.16 | 0.60 |
|  | 40–44 | 2.48 | 7.49 | 10.01 | 3.76 | 3.75 | 13.56 | 9.72 | 12.80 |
|  | 45–49 | 9.32 | 3.53 | 1.20 | 1.22 | 4.97 | 2.53 | 6.39 | -7.10 |
|  | 50–54 | 1.23 | 3.65 | 3.61 | 3.57 | 4.76 | 1.20 | 2.43 | -3.60 |
|  | 55–59 | 2.88 | 2.79 | 0.00 | 4.01 | 3.92 | 0.00 | 6.33 | 0.72 |
|  | 60–64 | 1.80 | 1.73 | 1.67 | 8.02 | 1.55 | 1.51 | 5.87 | 11.04 |
|  | 65–69 | 0.00 | 0.00 | 6.67 | 0.00 | 2.01 | 1.93 | 5.57 | **121.70***  |
|  | 70–74 | 0.00 | 0.00 | 0.00 | 0.00 | 0.00 | 2.71 | 0.00 | **81.10***  |
|  | 75–79 | 0.00 | 0.00 | 0.00 | 0.00 | 3.66 | 0.00 | 3.52 | **115.80***  |
|  | 80+ | 0.00 | 0.00 | 2.86 | 2.74 | 0.00 | 0.00 | 0.00 | -22.90 |
| **Skin color** |  |  |  |  |  |  |  |  |  |
|  | White | 4.37 | 4.04 | 3.82 | 3.21 | 4.26 | 5.02 | 6.65 | **6.40***  |
|  | Non-White | 8.25 | 4.34 | 10.54 | 7.60 | 10.41 | 9.93 | 13.21 | **11.10***  |
| **Motivation** |  |  |  |  |  |  |  |  |  |
|  | Gender-related | 1.98 | 0.82 | 1.22 | 1.45 | 1.36 | 1.84 | 1.25 | 2.34 |
|  | Family crime | 1.73 | 0.82 | 0.98 | 1.45 | 1.20 | 1.60 | 1.25 | 2.41 |
|  | Sex crime | 0.25 | 0.00 | 0.24 | 0.00 | 0.16 | 0.24 | 0.00 | 2.79 |
|  | Non-gender-related | 3.04 | 3.28 | 3.74 | 2.58 | 3.93 | 4.09 | 6.48 | **11.40***  |
|  | Robbery resulting in death | 0.16 | 0.33 | 0.24 | 0.32 | 0.48 | 0.32 | 0.72 | **16.96** |
|  | Criminal activity | 2.88 | 2.95 | 3.50 | 2.26 | 3.45 | 3.77 | 5.76 | **10.80***  |
|  | Men in POA | 67.89 | 74.53 | 96.77 | 76.58 | 98.67 | 102.66 | 121.73 | **10.70***  |
|  | Women in POA | 5.11 | 4.67 | 5.45 | 4.11 | 5.70 | 6.18 | 8.09 | **7.10***  |
|  | Women in POA—standardized | 4.98 | 5.17 | 5.39 | 4.06 | 6.05 | 6.29 | 10.85 | 10.69 |
|  | Women in Brazil, standardized | 5.60 | 5.48 | 5.53 | 5.49 | 5.48 | 5.30 | 5.25 | **-0.96***  |

*Indicates an average percent change (APC) significantly different from 0

were similar in men and women, especially for non-gender-related homicides. There is a debate in criminological research about the extent to which lethal violence differs according to gender-specific factors, with some groups advocating gender-sensitive approaches and others advocating gender-insensitive approaches [17]. The findings reported herein possibly contribute to the debate by demonstrating the importance of assessing the type of homicide. The phenomenon appears to be new: while gender-related homicides were stable and followed a recognized pattern, non-gender-related homicides had a volatility of a similar magnitude to that seen in male homicides.

The data collected in this study revealed a new and worrying situation in Brazil. We showed a growing number of homicides of women resulting from urban violence (crimes), from violent actions whose targets were men they knew, or from direct involvement in organized crime [7, 18, 19]. Since the 1980s, there has been a sharp increase in homicide rates for Brazilian women. A concomitant change has been observed in the deaths of less-educated, young,

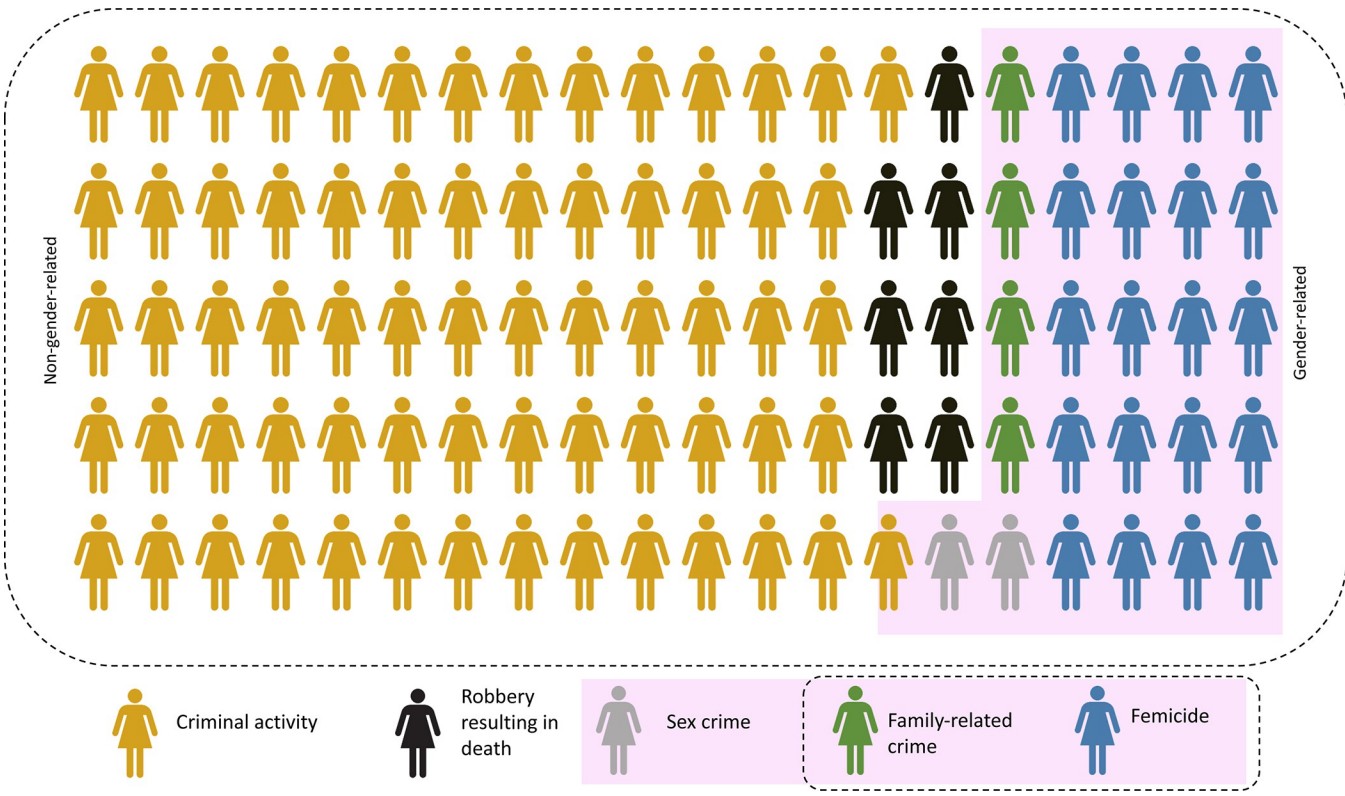

**Fig 1. Proportion of female homicide according to motivation (n = 459).**

Black men from the urban periphery with criminal records [20]. At the national level, there was a migration of crime from large urban centers to other regions of the country, where social inequality, access to firearms, drug trafficking, and alcohol and drug abuse contributed to Brazil presenting a mortality risk ten times higher than that of developed countries [5, 7]. In the study area, the increase in female homicides followed that of male homicides, with clearer trends in adolescents aged 15 to 19 years, predominantly those of non-White color and linked to criminal activities. In this sample, deaths related to criminal activity were associated with single, young, non-White women, and the vast majority were firearm victims with toxicology testing showing the presence of drugs. Psychotropic substances found in the victims' bodies can be considered a factor of vulnerability to victimization since the effects of intoxication reduce the notion of risks while drug use increases the exposure to predatory social environments.

These findings are in line with statistics in Brazil, where the homicide rate for Black women is 5.2 per 100,000 population while that of White women is 2.8. This difference has grown in the past decade: from 2007 to 2017, there was an increase of 29.9% in the homicide of Black people versus 4.5% in White people [21]. The main studies on the topic attribute this epidemiological finding to a historically increased risk of Black women suffering intimate partner violence [22]. Also, childhood sexual abuse has a higher incidence in Black girls, ranging from 14% to 44%, with a higher proportion of severe forms of violence such as oral, anal, or vaginal penetration [23]. Recent studies proposed that Black women who are homicide victims and perpetrators are part of the same population in socioeconomic, educational, and even behavioral or criminal terms [24], suggesting that other mechanisms are involved. This study highlights the often neglected role of crime, drug trafficking, and urban violence. An increase in

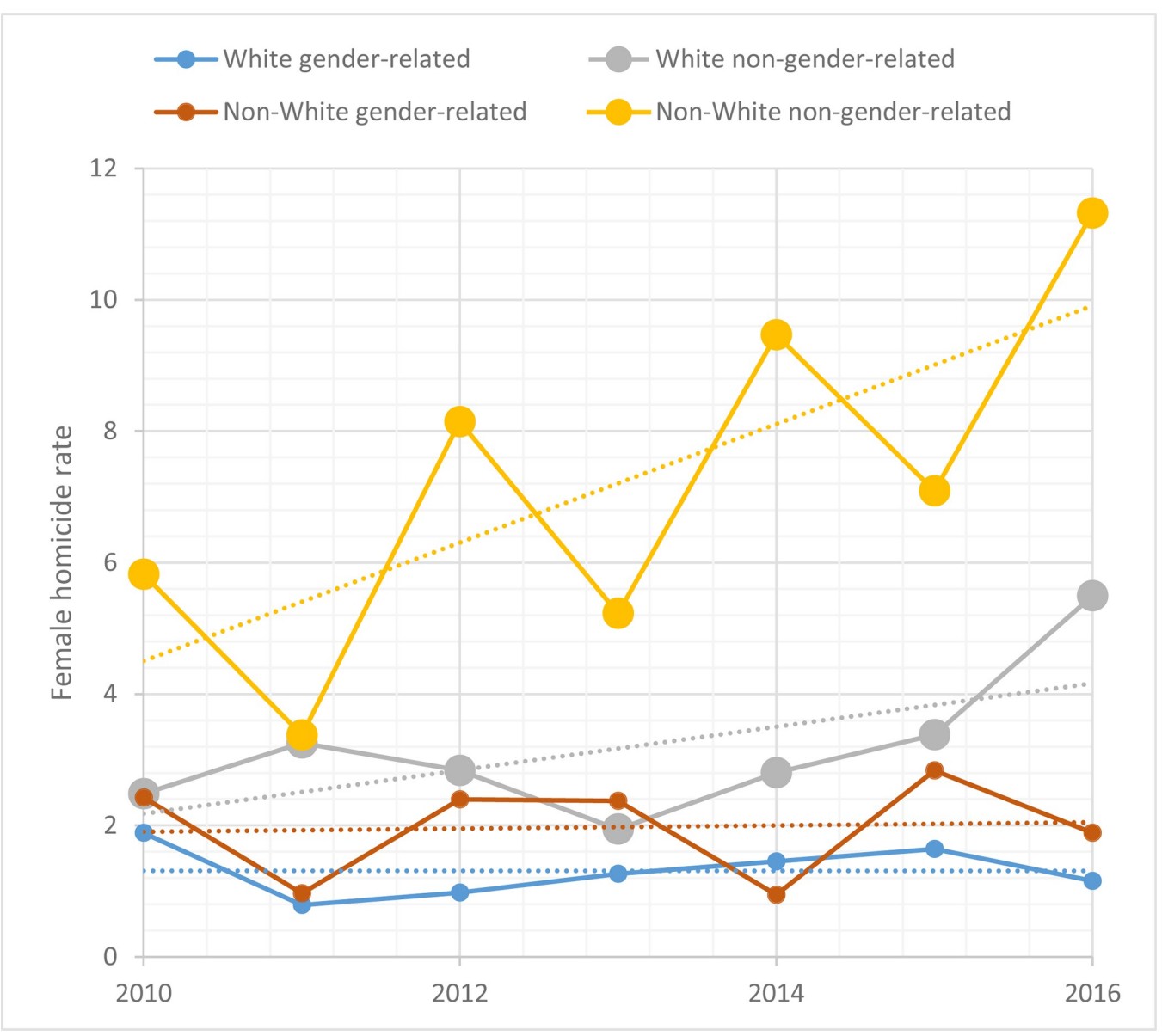

**Fig 2. Female homicide rate in Porto Alegre between 2010–2016 according to motivation and skin color.**

the number of non-White women in female homicides related to criminal activity is possibly linked to the greater exposure and socioeconomic vulnerability of this population, which is historical and results from persistent poverty.

The link between women and organized crime has been addressed by several fields of study using different methodologies. The respect and prestige attributed to male drug traffickers would be motivating factors for women to enter and stay in drug trafficking [25, 26]. The use of carefully collected medicolegal and police data can help separate gender-related homicides from those motivated by other violent acts. While women linked to crime are vulnerable to the violent environment, it is also possible that they are victims of confrontations with other criminals and the police, in addition to violence occurring in the domestic environment. In this study, we could not assess these differences, and this is a limitation to the interpretation of the data.

Researchers from different countries have linked female homicides to the availability and use of firearms [27–30]. In Brazil, there was a 592.8% increase in deaths caused by firearms from 1980 to 2014 [7–9]. Worldwide, firearms accounted for 42% of total homicides in 2010 and 54% of violent deaths in 2017 [31, 32]. In Italy, they accounted for 31% of violent female deaths from 2000 to 2005. In the United States, a study that analyzed homicides from 1981 to 2013 found a strong association between firearm ownership and the murder of women by intimate partners. American women were eleven times more likely to die from firearm injuries compared with those in other developed countries [33, 34]. The results of our study are in line with those of another Brazilian study that evaluated female homicides for 35 years and showed that firearms were the main instrument causing deaths, followed by sharp or blunt objects and asphyxia (strangulation) [7]. Here, the use of firearms was a strong predictor of non-gender-related homicide; it is one more factor in common between male and female homicide related to criminal activity.

Compared with international data, the prevalence of deaths related to urban violence in the study area is noteworthy. The involvement of young people in drug trafficking, other criminal activities, and interpersonal conflicts all increase the number of homicides in younger groups. Robbery resulting in death is also related to urban violence but was classified separately in the present study because a distinct trend in the victims of this crime was observed. This crime generally involved an older group compared with other groups of crime victims [35]. Robbery resulting in death reflects several complex social conditions, including stealing objects to easily trade for drugs, unplanned growth of the urban population, and large circulation of weapons [36].

In this study, the second cause of violent deaths among women was femicide. Intimate partner violence is a widespread public health problem in countries of the Americas. In the United States, a study involving 18 states showed that 55.3% of female homicide victims were murdered by their intimate partners versus only 5% of men [37]. A study of 24 countries in the region revealed that this type of violence affects 14% to 60% of the female population of childbearing age (15–49 years). Other studies have shown a relationship between increased urban violence and elevated gender violence, especially where great socioeconomic inequality and the influence of organized crime and drug trafficking exist [5, 7, 38]. In Brazil, the prevalence is that one in seven women (14%-17%) may be a victim of intimate partner violence at some point in life [4]. Interestingly, a recent analysis of Brazilian administrative data shows an association between gender wage ratio and interpersonal violence against women, including homicide [39]. In that specific dataset, larger gender gaps were associated with more interpersonal violence. This may lend support to the ameliorative hypothesis, that is, that gains in gender equality generates more equitable responsibility and men have less power and authority over women. Since raw wage ratios appear stable in a period similar to this study and we look into only one region, and as we do not have access to direct data on family wages, we cannot analyze gender gap as a factor contributing to homicide rates in this report.

One hypothesis for increased female homicides in younger groups is the abuse of alcohol and other substances by the younger generation of men, which makes them victims and perpetrators of multiple forms of violence, including gender-based violence [9, 40]. Lethal domestic violence implies a significant loss of life years and a reduction in overall life expectancy in affected families [41]. In contrast to the results for non-gender-related deaths, there were no significant findings of the presence of alcohol or drugs in the blood and urine samples collected from the victims.

The study has common limitations arising from the use of secondary data. Missing data in public databases include the identification of ignored values or blank data, especially information on victims' education, income, and marital status. The hypothesis of the type of homicide

(gender-related or not) was based on immediate police and autopsy data, and we did not perform extensive psychometric testing on the instrument used to classify the cases. Although the investigation and trial of the crime could provide a more definitive classification and other relevant information, these data were not generally available at data collection—the average court processing time for homicide cases in Brazil is over 8 years [42]. The geographic coverage was also limited to one state in South Brazil. As male cases were not the focus of this study, the types of homicide were not recorded, only the rates for reference purposes, which is a limitation for understanding the phenomenon in men and making more specific comparisons. Nonetheless, the combination of medicolegal and police data allowed us to demonstrate a relevant phenomenon: the increase in female homicide rates due to non-gender-related factors, especially criminal activities.

Integrating different data sources on female homicides can lead to a better understanding of how these crimes occur. We present herein a worrying phenomenon, ie, the increase in female homicide rates due to non-gender-related factors, generally characterized as urban violence, which accounted for over 70% of the homicides in the study area in the period. Increases in non-gender-related homicide affected young women and adolescent girls when characterized as linked to criminal activity, and older women when characterized as robbery resulting in death. Rates were also higher in non-White women, especially for non-gender-related homicides. Knowledge of these characteristics should be used to guide public policies through the engagement of public and private institutions dedicated to health protection, public safety, and education. Providing specialized services for the protection of women and girls in vulnerable situations and sharing clear and up-to-date information will contribute to reducing the risk of death.

## Author Contributions

**Conceptualization:** Angelita Maria Ferreira Machado Rios, Lisieux Elaine De Borba Telles, Pedro V. S. Magalhães.

**Data curation:** Angelita Maria Ferreira Machado Rios, Kleber Cardoso Crespo, Murilo Martini, Lisieux Elaine De Borba Telles.

**Formal analysis:** Angelita Maria Ferreira Machado Rios, Murilo Martini, Pedro V. S. Magalhães.

**Investigation:** Kleber Cardoso Crespo, Lisieux Elaine De Borba Telles, Pedro V. S. Magalhães.

**Methodology:** Kleber Cardoso Crespo, Murilo Martini, Lisieux Elaine De Borba Telles, Pedro V. S. Magalhães.

**Project administration:** Lisieux Elaine De Borba Telles, Pedro V. S. Magalhães.

**Supervision:** Pedro V. S. Magalhães.

**Writing – original draft:** Angelita Maria Ferreira Machado Rios, Lisieux Elaine De Borba Telles, Pedro V. S. Magalhães.

**Writing – review & editing:** Angelita Maria Ferreira Machado Rios, Kleber Cardoso Crespo, Murilo Martini, Lisieux Elaine De Borba Telles, Pedro V. S. Magalhães.

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
