## [Decision Letter · Decision Letter 0]

12 Jul 2022

PONE-D-21-38852Gender-related and non-gender-related female homicide in Porto Alegre, Brazil, from 2010 to 2016PLOS ONE

Dear Dr. Magalhaes,

Thank you for submitting your manuscript to PLOS ONE. After careful consideration, we feel that it has merit but does not fully meet PLOS ONE’s publication criteria as it currently stands. Therefore, we invite you to submit a revised version of the manuscript that addresses the points raised during the review process.

We look forward to receiving your revised manuscript.

Kind regards,

Vanessa Carels

Staff Editor

PLOS ONE

Journal Requirements:

2. In ethics statement in the manuscript and in the online submission form, please provide additional information about the patient records used in your retrospective study. Specifically, please ensure that you have discussed whether all data were fully anonymized before you accessed them and/or whether the IRB or ethics committee waived the requirement for informed consent (for instance by the parents of the deceased minors or next of kin, if applicable)

Reviewers' comments:

Reviewer's Responses to Questions

**Comments to the Author**

1. Is the manuscript technically sound, and do the data support the conclusions?

Reviewer #1: Yes

Reviewer #2: Yes

2. Has the statistical analysis been performed appropriately and rigorously? 

Reviewer #1: I Don't Know

Reviewer #2: Yes

3. Have the authors made all data underlying the findings in their manuscript fully available?

Reviewer #1: No

Reviewer #2: No

4. Is the manuscript presented in an intelligible fashion and written in standard English?

Reviewer #1: Yes

Reviewer #2: Yes

5. Review Comments to the Author

Reviewer #1: Congratulations for your work!

The manuscript deals with a problem of national interest.

Data from the morgue of the Medicolegal Department of Porto Alegre, Brazil, from 2010 to 2016, was used to analyze female homicide.

Introduction

The Introduction addresses the research problem.

In the last paragraph: the aim presented did not represent all the manuscript´s objectives.

I would suggest showing more clearly what is the originality of its contribution and the hypotheses behind the study.

Methods

Could you explain why did you use Joinpoint regression? Population data has many autocorrelation problems. This method is appropriate to deal with this problem? Also, you have a small series, with data from seven years. Joinpoint regression is the better choice for that?

Could you explain which source of the population did you use to build the homicide rates according to each age group, skin color, and type of homicide? Censo 2010? IBGE projection or another source?

You mentioned performing logistic regression. Could you present what variables (dependent and independent) were used and what procedures and tests did you do?

Results

Table 1: For me, it seems to be a Figure.

Table 2: Could you improve the table name, including information about the person, place, and time? Also, include data source.

Table 3: Could you improve the table name, including information about the person, place, and time? Also, include data source and the 95% Interval confidence of AAPC.

I did not find the table with results from logistic regression (intermediate and/or final models, variables used, OR with 95% IC and p-value).

Reviewer #2: This is a well-written piece of work describing female homicide victimisation in Brazil. Female homicide victimisation is frequently equated with femicide. This article could be a welcome addition to the literature, as it challenges our common perception of women mostly being killed in domestic contexts - criminal contexts and other contexts are oftentimes ignored. In order to do so, the authors should make very clear what they consider "femicide", and "gender-related" versus "non-gender-related" (and how they coded a case as gender-related versus non-gender related). This distinction and operationalization is not clear, as different definitions are used throughout the text.

In addition to this main aspect, please find below several comments that may help clarify / improve the paper.

1. It may be helpful to briefly refer to the amelioration / backlash hypotheses in the introduction.

2. Also, please provide some background information on gender equality measures in Brazil (and perhaps how this has changed over time -- this could be a valuable aspect in explaining the increase in female homicide victimisation rates, linking to comment #1 above).

3. On p.7, more information is needed on how cases were coded -- was a validated instrument used? Was an instrument created, based on the UNODC classification? What was done when there was no consensus in coding? What was the interrater reliability?

4. The inclusion of immolation as a m.o. may generate false positives due to a ver small cell count (perhaps merge with another similar category).

5. Please alos present the total N per category in table 1.

6. I would like to see Table 3 presented as a graph, since much of the article is focused around trends -- a graph would work better to provide an overview of such trends.

7. The discussion should be expanded by going deeper into gendered versus non-gendered related factors. This also ties in with my main point of concern outlined above: A clear description of what can be understood as gender-related violence is missing.

8. In the discussion, please elaborate on how we can explain the role of firearms in female homicide victimisation, and how this may be similar / different from male victimisation.

9. It may be helpful to know how the authors came up with this final sample of cases - what about the flow of such cases through the system (see Ludmila Ribeiro's work on this): How many cases were originally identified? How many turned out to be suicides or accidents instead? How many were solved, etc. This provides the reader with a better perspective of the validity of this sample.

6. PLOS authors have the option to publish the peer review history of their article (what does this mean?). If published, this will include your full peer review and any attached files.

Reviewer #1: No

Reviewer #2: **Yes: **Marieke Liem

---

## [Author Response · Author response to Decision Letter 0]

23 Sep 2022

Dear Dr. Carels

We appreciate the opportunity to submit a revised version of our manuscript “Gender-related and non-gender-related female homicide in Porto Alegre, Brazil, from 2010 to 2016” for consideration for publication in PLOS ONE. As recommended, we address the reviewers’ concerns on an item-by-item basis and highlight changes. We hope that the current version of the manuscript satisfies the reviewers and the editor.

Sincerely, 

Prof. Pedro VS Magalhães 

Universidade Federal do Rio Grande do Sul - UFRGS 

Rua Ramiro Barcelos, 2350 

Porto Alegre/RS Brasil 90035-903 

Phone: + 55 51 3359 8845 

Fax: + 55 51 3359 8846 

E-mail address: pedromaga2@gmail.com

Reviewer #1

1. In the last paragraph: the aim presented did not represent all the manuscript´s objectives. I would suggest showing more clearly what is the originality of its contribution and the hypotheses behind the study.

Response: We add a sentence to the paragraph's end to clarify the aims of the study.

We report global changes in homicide rates in women in the period, as well as changes in relevant subgroups according to age, race and motivation for the homicide.

2. Could you explain why did you use Joinpoint regression? Population data has many autocorrelation problems. This method is appropriate to deal with this problem? Also, you have a small series, with data from seven years. Joinpoint regression is the better choice for that?

Response: Joinpoint regression is an increasingly popular method for analyzing trends in time series data as it can help identify calendar years in which the temporal trend changed significantly. The method assumes that data can be divided into subsets with unique linear trends. A joinpoint is a point in time when the population parameters change. We add these arguments to the analysis section of the manuscript. Joinpoint regression can deal with autocorrelated data in time series, such as the data presented here (see Kim et al, (2000)). There are other methods for dealing with such data, but joinpoint regression has proved a reliable method. We add some more explanation to the analysis section. 

Joinpoint regression is an increasingly popular method for analyzing trends in time series data as it can help identify calendar years in which the temporal trend changed significantly. The method assumes that data can be divided into subsets with unique linear trends. A joinpoint is a point in time when the population parameters change. Joinpoint regression joins multiple straight lines in logarithmic scale in order to detect the annual trend. The analysis starts with the minimum number of inflections to assess whether one or more joinpoints are statistically significant and whether they should be added to the model. For the analysis of trends, we sought to identify the regression equation that best described the relationship between the independent variable (year) and the dependent variable (homicide rates). We calculated rates and standard errors for each year for each group of interest and thus obtained the annual percent change (APC) for each group. 

3. Could you explain which source of the population did you use to build the homicide rates according to each age group, skin color, and type of homicide? Censo 2010? IBGE projection or another source?

Response: All homicide data were collected from autopsy reports issued by the Porto Alegre medicolegal department, to which the authors were granted access. Population data from the national census were used to construct homicide rates. Resident population, stratified by sex, age, year and skin color were obtained from the Brazilian Institute of Geography and Statistics (IBGE). For the calculation of age-standardized rates, the number of deaths in the analyzed year was considered as the numerator and the estimated population of the respective year was considered as the denominator. 

This retrospective study of female homicides occurring from January 2010 to December 2016 was based on an analysis of autopsy reports issued by the Porto Alegre Medicolegal Department. The estimated average population covered by the morgue in the study period was 2,334,730 people, including the population of the Rio Grande do Sul state capital, Porto Alegre (60.8%), and nine additional cities in the metropolitan area. 

Resident population, stratified by sex, age, year and skin color were obtained from the Brazilian Institute of Geography and Statistics (IBGE). For the calculation of age-standardized rates, the number of deaths in the analyzed year was considered as the numerator and the estimated population of the respective year was considered as the denominator.

4. You mentioned performing logistic regression. Could you present what variables (dependent and independent) were used and what procedures and tests did you do?

Response: We add detail to the logistic regression analysis and results in this revised version.

A logistic regression model constructed a priori and consisting of age, skin color, marital status, previous occurrences, place of death, the occurrence of other deaths in the same place, the instrument used, single injury, and toxicology was used to predict gender-related homicides.

In a logistic regression model (AUC 0.84, Nagelkerke R2 = 0.42), significant predictors of gender-related homicide were age (OR 0.97, 95% CI 0.94-0.99, p=0.011), death at home (OR 7.70, 95% CI 3.88-15.30, p<0.001), single injury (OR 6.46, 95% CI 1.22-4.57, p=0.011), no drugs of abuse in the toxicology test (OR 3.20, 95% CI 1.39-7.37, p=0.006), and use of methods other than firearms (OR 0.14, 95% CI 0.07-0.29, p<0.001). Hosmer & Lemeshow test indicated a good fit for the model (p=0.247).

5. Table 1: For me, it seems to be a Figure.

Response: Table 1 only reports on definitions used in the study.

6. Table 2: Could you improve the table name, including information about the person, place, and time? Also, include data source.

Response: We edited the table legend for clarity.

Table 2. Characteristics of homicide victims from the Porto Alegre Medicolegal Department from January 2010 to December 2016 according to motive (n=459)

7. Table 3: Could you improve the table name, including information about the person, place, and time? Also, include data source and the 95% Interval confidence of AAPC.

Response: We edited the table legend for clarity

Table 3. Female homicide rates from the Porto Alegre Medicolegal Department from January 2010 to December 2016 per 100,000 population

8. I did not find the table with results from logistic regression (intermediate and/or final models, variables used, OR with 95% IC and p-value).

Response: We used only one model constructed a priori to avoid overfitting. Results are shown as text (see above).

Reviewer #2

1. This article could be a welcome addition to the literature, as it challenges our common perception of women mostly being killed in domestic contexts - criminal contexts and other contexts are oftentimes ignored. In order to do so, the authors should make very clear what they consider "femicide", and "gender-related" versus "non-gender-related" (and how they coded a case as gender-related versus non-gender related). 

Response: the methods section (and table 1) include a discussion of femicide and gender-related versus non-gender-related homicide, which we attempt to make clearer in this revision. 

Homicides were then classified as those perpetrated by an intimate partner (femicide), by another family member (family-related homicide), sex crimes, homicides related to criminal activities (eg, drug trafficking and other crimes), and robbery resulting in death. The cases could then be as gender-related homicides (family-related including femicide and sex crimes) and non-gender-related homicides (robbery resulting in death and those related to criminal activity) to follow the United Nations Office on Drugs and Crime (UNODC) classification (2018) and to compare them with international data (see Table 1).

2. It may be helpful to briefly refer to the amelioration / backlash hypotheses in the introduction.

3. Please provide some background information on gender equality measures in Brazil (and perhaps how this has changed over time -- this could be a valuable aspect in explaining the increase in female homicide victimisation rates, linking to comment #1 above).

Response to 2 and 3. A add a comment on the wage gap and the ameliorative / backlash hypotheses.

Interestingly, a recent analysis of Brazilian administrative data shows an association between gender wage ratio and interpersonal violence against women, including homicide. In that specific dataset, larger gender gaps were associated with more interpersonal violence. This may lend support to the ameliorative hypothesis, that is, that gains in gender equality generate more equitable responsibility and men have less power and authority over women. Since raw wage ratios appear stable in a period similar to this study and we look into only one region, and as we do not have access to direct data on family wages, we cannot analyze gender gap as a factor contributing to homicide rates in this report.

4. On p.7, more information is needed on how cases were coded -- was a validated instrument used? Was an instrument created, based on the UNODC classification? What was done when there was no consensus in coding? What was the interrater reliability?

Response: the methods section includes an account of how cases were classified according to motivation. We attempt to make it clearer in this section. We did use an instrument based on UNODC classification, although we did not attempt to measure interrater reliability. This is added as a limitation of the study. 

Sociodemographic and criminal information was obtained from the Forensic Institute database, police reports, and records collected at the crime scene by the removal team; data regarding prior victimization were obtained from existing prior records of the Forensic Institute. Medicolegal variables were extracted from autopsy reports and forensic laboratory test results. Skin color (classified as White or non-White) was ascertained on postmortem examination, together with the number of injuries and causative agent.

The hypothesis of the type of homicide (gender-related or not) was based on immediate police and autopsy data, and we did not perform extensive psychometric testing on the instrument used to classify the cases.

5. The inclusion of immolation as a m.o. may generate false positives due to a ver small cell count (perhaps merge with another similar category).

Response: Immolation as a method of homicide is only presented in tabular form, so the reader can get a complete picture of all homicides. It is not used in any analyses.

6. Please alos present the total N per category in table 1.

Response: We add sample size per motivation to table 2. 

7. I would like to see Table 3 presented as a graph, since much of the article is focused around trends -- a graph would work better to provide an overview of such trends.

Response: We would like to keep the table. The problem with using a graphic representation (we made several versions) is that the data contains many subcategories, and that makes it harder to understand the trends. The table better accommodates the data, in our view.

8. The discussion should be expanded by going deeper into gendered versus non-gendered related factors. This also ties in with my main point of concern outlined above: A clear description of what can be understood as gender-related violence is missing.

Response: Our definitions are tied to the UNODC definition of homicide related to gender and not related to gender. This is explained in Table 1 and we add a note to Table 2. 

The UNODC in “Gender-related killing of women and girls” states that “The notion of gender-related killing (...) is subject to a certain degree of interpretation. Nevertheless, some aspects of gender-related killing of women are indisputable, one being that this type of homicide is part of female homicide, yet not all female homicides are gender related. (...) In broader terms, such killings can be divided into those perpetrated within the family and those perpetrated outside the family sphere. Gender-related killing of women and girls is analysed in this study using the indicator for intimate partner/family-related homicide. This provides a concept that covers most gender-related killings of women, is comparable and can be aggregated at global level”. We employ this definition here, and, by contrast, all homicides that fall outside this category are labelled non-gender related.

9. In the discussion, please elaborate on how we can explain the role of firearms in female homicide victimisation, and how this may be similar / different from male victimisation.

Response: In this data, the use of firearms was one the strongest indicators of a non-gender-related homicide. The use of firearms in criminal activity is a common pattern in Brazil, and we believe it should be taken as a similarity between male and female homicide, reinforcing the trend shown of female homicide mirroring male suicide. We add a comment in the discussion section. 

Here, the use of firearms was a strong predictor of non-gender-related homicide; it is one more factor in common between male and female homicide related to criminal activity.

10. It may be helpful to know how the authors came up with this final sample of cases - what about the flow of such cases through the system (see Ludmila Ribeiro's work on this): How many cases were originally identified? How many turned out to be suicides or accidents instead? How many were solved, etc. This provides the reader with a better perspective of the validity of this sample.

Response: The point is well taken, but as we expose in the discussion section, investigation and trial data is not available for the sample, as it takes an average court processing time of over 8 years in Brazil for homicides. This is a limitation of the data and is presented as such.

---

## [Editor Report · Decision Letter 1]

5 Feb 2023

Gender-related and non-gender-related female homicide in Porto Alegre, Brazil, from 2010 to 2016

PONE-D-21-38852R1

Dear Dr. Magalhaes,

We’re pleased to inform you that your manuscript has been judged scientifically suitable for publication and will be formally accepted for publication once it meets all outstanding technical requirements.

Kind regards,

Senthil Kumaran, MBBS, MD, DNB

Academic Editor

PLOS ONE
---

## [Editor Report · Acceptance letter]

6 Mar 2023

PONE-D-21-38852R1 

Gender-related and non-gender-related female homicide in Porto Alegre, Brazil, from 2010 to 2016 

Dear Dr. Magalhães:

I'm pleased to inform you that your manuscript has been deemed suitable for publication in PLOS ONE. Congratulations! Your manuscript is now with our production department. 

Kind regards, 

on behalf of

Dr. Senthil Kumaran 

Academic Editor

PLOS ONE